# Phytostabilization of Zn and Cd in Mine Soil Using Corn in Combination with Biochars and Manure-Based Compost

**Gilbert C. Sigua [1,\*], Jeff M. Novak [1], Don W. Watts [1], Jim A. Ippolito [2], Thomas F. Ducey [1], Mark G. Johnson [3] and Kurt A. Spokas [4]**

[1] Department of Agriculture, Agricultural Research Service, Coastal Plains Soil, Water, and Plant Research Center, Florence, CA 29501, USA; jeff.novak@ars.usda.gov (J.M.N.); don.watts@ars.usda.gov (D.W.W.); tom.ducey@ars.usda.gov (T.F.D.)

[2] Department of Soil and Crop Sciences, C006 Plant Sciences Building, Colorado State University, Fort Collins, CO 80523, USA; jim.ippolito@colostate.edu

[3] United States Environmental Protection Agency, National Health and Environmental Effects Research Laboratory, Western Ecology Division, 200 Southwest 35th Street, Corvallis, OR 97333, USA; johnson.markg@epa.gov

[4] Department of Agriculture, Agricultural Research Service, St. Paul, MN 55108, USA; kurt.spokas@ars.usda.gov

\* Correspondence: gilbert.sigua@ars.usda.gov; Tel.: +1-843-669-5203

**Abstract:** Mining activities could produce a large volume of spoils, waste rocks, and tailings, which are usually deposited at the surface and become a source of metal pollution. Phytostabilization of the mine spoils could limit the spread of these heavy metals. Phytostabilization can be enhanced by using soil amendments such as manure-based biochars capable of immobilizing metal(loid)s when combined with plant species that are tolerant of high levels of contaminants while simultaneously improving properties of mine soils. However, the use of manure-based biochars and other organic amendments for mine spoil remediation are still unclear. In this greenhouse study, we evaluated the interactive effect of biochar additions (BA) with or without the manure-based compost (MBC) on the shoots biomass (SBY), roots biomass (RBY), uptake, and bioconcentration factor (BCF) of Zn and Cd in corn (*Zea mays* L.) grown in mine soil. Biochar additions consisting of beef cattle manure (BCM); poultry litter (PL); and lodge pole pine (LPP) were applied at 0, 2.5, and 5.0% (w/w) in combination with different rates (0, 2.5, and 5.0%, w/w) of MBC, respectively. Shoots and roots uptake of Cd and Zn were significantly affected by BA, MBC, and the interaction of BA and MBC. Corn plants that received 2.5% PL and 2.5% BCM had the greatest Cd and Zn shoot uptake, respectively. Corn plants with 5% BCM had the greatest Cd and Zn root uptake. When averaged across BA, the greatest BCF for Cd in the shoot of 92.3 was from the application of BCM and the least BCF was from the application of PL (72.8). Our results suggest that the incorporation of biochar enhanced phytostabilization of Cd and Zn with concentrations of water-soluble Cd and Zn lowest in soils amended with manure-based biochars while improving the biomass productivity of corn. Overall, the phytostabilization technique and biochar additions have the potential to be combined in the remediation of heavy metals polluted soils.

**Keywords:** biochar; phytoextraction; corn; uptake; mine soils; heavy metals; root biomass; shoot biomass

## 1. Introduction

Mining activities usually produce a large volume of spoils, waste rocks, and tailings, which are usually deposited at the soil surface. If the spoils contain heavy metals that are soluble, there is

a potential of heavy metal pollution contamination and off-site movement. Mined areas near Webb City in Jasper County, Missouri, contained mine waste piles that were removed, but still provide a source of heavy metal contamination, particularly Zn and Cd in the underlying soil. Mining activities can lead to extensive environmental pollution of terrestrial ecosystem due to the deposition of heavy-metal containing waste materials, tailings, and lagoon wastes [1–3].

Metal (loid) contaminants such as Cd and Zn are significant issues, not only for the environment, but especially for human health [4–6]. These contaminated areas present a health risk and are recognized as areas that need to be remediated to allow for crop phytostabilization to occur [1]. Often, contaminated sites are not conducive for plant growth due to metal toxicity, lack of soil nutrients, low pH values, poor microbial activity, and unsuitable physical soil properties. Both physical and chemical techniques have been considered in mine spoil remediation, but these methods have flaws, are expensive, and can be disruptive to soils. Remediation of these contaminated and hazardous soils by conventional practices using excavation and landfilling is arguably unfeasible on large scales because these techniques are cost-prohibitive and environmentally disruptive [7,8]. Phytostabilization techniques that involve the establishment of plant cover on the surface of contaminated sites could serve as an efficient alternative remediation approach as they provide low-cost and environmentally friendly options [7,9]. For this reason, remediation of contaminated sites using phytostabilization techniques require the amendment to improve soil-plant relationships thereby stimulating plant growth.

Remediation of mine spoil can be a complex process due to several chemical and physical factors that can limit plant growth [10]. Bolan et al. [11] summarized the different factors affecting phytostabilization. For example, soil, plant, contaminants, and environmental factors determine the successful outcome of phytostabilization technology in relation to both the remediation and revegetation of contaminated sites. Mine spoils may have unfavorable soil chemical characteristics, e.g., very low pH, phytotoxic metals [12,13], physical limitations (e.g., high bulk density, low soil moisture retention, poor aggregation [14]; and unsuitable microbial habitat conditions, e.g., low soil organic matter and poor nutrient turnover [15]. These aspects can severely limit plant growth. As such, reclamation plans usually involve applying soil amendments (i.e., composts, lime) to neutralize their low pH, and to raise organic matter levels that favors organic binding of metals, along with enhanced microbial enzymatic activity for nutrient cycling [16].

Phytostabilization can be enhanced by using soil amendments that immobilize metal(loid)s when combined with plant species that are tolerant of high levels of contaminants while simultaneously improving the physical, chemical, and biological properties of mine soils. Some previously used amendments to improve soil conditions include biosolids, lime, green waste, or biochars. Among these amendment types, the use of biochar has recently been investigated for in situ remediation of contaminated lands in association with plants [10,17–19]. The incorporation of organic amendments improves the quality of mine soils and makes it possible for vegetation to be established [20,21]. Recent studies have highlighted that biochars are effective soil amendments in that they improve soil conditions to raise the agronomic values of soils [22–25].

Numerous studies may have shown that adding organic amendments (e.g., biochars, sewage sludge, manures) to soil promotes the phytoextraction process [26,27], but only few studies have evaluated the combined effect of organic amendments and phytostabilization with corn in Cd and Zn contaminated mine soils. There is a lack of agreement over the influence of organic amendments such as biochars on metal immobilization in soil. Moreover, the application of biochars to contaminated soil systems has not been systematically investigated to any great extent. Biochar may be a tool for mine spoil remediation; however, its mechanisms for achieving this goal are still not well understood. The objective of our study was to evaluate the interactive effects of biochar additions with or without the manure-based compost (MBC) on shoots biomass (SBY), roots biomass (RBY), uptake, and bioconcentration factor (BCF) of Zn and Cd in corn (*Zea mays* L.) grown in mine soil.

## 2. Materials and Methods

### 2.1. Site Description, Soil Characterization, and Soil Preparation

A field for sampling soil was selected near Webb City in Jasper County, MO (latitude 37.13°, longitude 94.45°). This location is a part of the Oronogo-Duenweg mining area of Southwest MO. Mining of lead (Pb) and zinc (Zn) ore has occurred across the country with leftover milling waste discarded in chat piles. The chat piles contain residual Pb and Zn concentrations that in some locations moved into the underlying soil.

Prior to the mining disturbance, soil in this field was mapped as a Rueter series, which is classified using United States Department of Agriculture Taxonomic terminology as a loamy-skeletal, siliceous, active, mesic Typic Paleudalf. Examination of the Reuter soil profile reveals that it has extremely gravelly silt loam textured soil horizons that formed in colluvium over residuum derived from limestone (Soil Survey of Jasper County, MO, 2002).

For our purposes, a backhoe was used to collect a few hundred kg of C horizon material down from 60- to 90-cm deep. The soil along with coarse fragments was placed in plastic-lined metal drums and transported to the ARS-Florence (Florence, SC, USA). The C horizon material was removed from the drums and air-dried. As a result of the presence of large cobbles, the soil was screened using a 12.7-mm diameter sieve to collect soil material more appropriate for use in a potted greenhouse experiment. Sieving the soil revealed that it contained approximately 30% (w/w) coarse fragments that were >12.7-mm in diameter. Soil that passed through the sieve was stored in the plastic line drums for characterization and used in our greenhouse experiment.

The sieve C horizon material (<12.7-cm diameter) was characterized for its pH (4.40) using a 1:2 (w/w) soil:deionized water ratio [16]. Additionally, bioavailable metal and total metal concentrations were extracted using multiple extractants and acid digestion, respectively. Both deionized water (water-soluble) and 0.01M $CaCl_2$ (extractable) metal concentrations were determined in triplicate by extracting 30 g soil with 60 mL of liquid extractant, shaken for 30 m, and filtered using a nylon 0.45 μM filter syringe [10,16,28]. Extraction with diethylenetriamine pentaacetic acid (DTPA) was also conducted in triplicate using 10 g of soil with 20 mL of DTPA after shaking for 2 h, and filtration using 0.45 μm filter syringe [10,16]. Total metal concentrations were determined in triplicate by digestion of 10 g soil in 100 mL of 4 M $HNO_3$ as described [28]. All water-soluble and extractable metal concentrations including Cd and Zn were quantified via the inductively coupled plasma spectroscopy atomic emission spectroscopy (ICP-AES) (Thermo Fisher Scientific, West Palm Beach, FL, USA). Concentrations of Cd and Zn and other chemical properties of C horizon are presented in Table 1.

**Table 1.** Chemical properties of Tri-State Mine soil (C horizon) used in the study.

| | | Extractable Metal (mg kg$^{-1}$) | |
| --- | --- | --- | --- |
| Element | Total Metal (mg kg$^{-1}$) [†] | H$_2$O | 0.01 M CaCl$_2$ |
| Al | ND | 4.40 ± 3.87 | 11.36 (1.56) |
| Cl | ND | ND | ND |
| Cd | 72.2 (2.7) | 5.73 ± 0.98 | 50.45 (0.40) |
| Cr | ND | 0 | 0.12 (0.01) |
| Cu | 66.5 (2.5) | 0.22 (0.07) | 2.17 (0.02) |
| Fe | ND | 10.57 (2.11) | 12.65 (1.45) |
| K | 711 (25) | 26.18 (3.47) | 59.48 (3.35) |
| Mg | 355 (45) | 4.53 (1.38) | 36.49 (0.97) |
| Mn | 72 (5.7) | 2.48 (0.72) | 21 (0.9) |
| Na | ND | 22.25 (4.01) | 25.58 (4.24) |
| Ni | 7.6 (0.3) | 0.18 (0.01) | 0.45 (0.01) |
| P | 168 (4) | 3.89 (0.06) | 1.43 (1.30) |
| Pb | 23.5 (0.7) | 0 | 0 |
| SO$_4$ | ND | 152.6 (19.4) | 112.8 (17.4) |
| Zn | 2225 (12) | 141.0 (25.7) | 782 (13) |

[†] samples digested using 4M HNO$_3$; (means of n = 3; standard deviation in parentheses; ND = not determined; 0 value = below detection limit).

## 2.2. Experimental Setup and Design

The experimental treatments consisted of biochar additions (BA): Beef cattle manure (BCM); poultry litter (PL); and lodge pole pine (LPP) that were applied at 0, 2.5, and 5.0% (w/w) in combination with different rates (0, 2.5, and 5.0%, w/w) of MBC (RMBC), respectively. Experimental treatments were replicated three times using a 3 × 2 × 3 split plot arrangement in completely randomized block design.

The treated and untreated C material soils were placed into triplicate plastic flower pots (15-cm top diameter × 17-cm deep) and gently tapped to a bulk density of 1.5 g/cm$^3$ as outlined in Novak et al. [16]. Eight corn seeds were then planted in each pot. The pots were transported to a greenhouse and randomly placed on benches. Corn in the pots were kept in the greenhouse under a mean air temperature of about 21.8 ± 3.1 °C and relative humidity of about 53 ± 12.2%. On day 16, all pots were fertilized with a 10 mL solution of NH$_4$NO$_3$ that delivered an equivalent of 25 kg N ha-1 because some treatments exhibited N deficient response in corn leaves (yellowing). No inorganic P or K was added to the pots because these nutrients were supplied with the amendments. The pots were watered by hand using recycled water several times per week.

## 2.3. Feedstock Collection, Description, Biochar Production, and Characterization

Three feedstocks were used to produce biochars in this experiment namely: Beef cattle manure; lodge pole pine; and poultry litter. The raw beef cattle manure was collected from a local feedlot operation near Webb City, MO. The manure pile was exposed to the environment for 1–2 years to allow for conversion into a manure-based compost mixture. A few kg of the manure compost was transported to the ARS-Florence location and sieved using a 6-mm sieve. A portion of the 6-mm sieved manure compost was pyrolyzed at 500 °C into biochar as outlined in Novak et al. [29]. The remaining two biochars were available commercially and consisted of biochar produced from the poultry litter and lodgepole pine feedstocks. The poultry litter biochar was produced by gasification using a fixed-bed pyrolyzer and the lodgepole pine biochar was produced using a slow pyrolysis process. The pyrolysis temperatures employed to produce these two biochars are not available.

All three biochars were characterized for their pH and electrical conductivity in a 1:2 (w/w) biochar to deionized water ratio [16]. All three biochars were also characterized chemically (ASTM D3176; Hazen Research, Inc., Golden, CO, USA). The molar H/C and O/C ratios were calculated from the elemental analysis. Total elemental composition of all three biochars was determined using concentrated HNO$_3$ acid digestion described in the US EPA 305b method [29,30] and were quantified using an inductively coupled plasma atomic emission spectroscopy (ICP-AES). Similar

characterization was performed on the beef cattle manure compost feedstock as described above. Some of the chemical and physical properties of the manure-based compost and biochars are shown in Table 2. The appropriateness of using the different designer biochars in our study were based on an early published paper by Novak et al. [16].

**Table 2.** Chemical and physical properties of compost and biochars (dry-basis).

| A. Ultimate and Proximate Analysis | | | | |
|---|---|---|---|---|
| | **Beef Cattle Manure** | | **Lodgepole Pine** | **Poultry Litter** |
| **Measurement (%)** | **Compost** | **Biochar** | **Biochar** | **Biochar** |
| C | 17.5 | 13.8 | 90.5 | 37.4 |
| H | 1.9 | 0.7 | 2.4 | 2.8 |
| O | 10.5 | 1.4 | 3.2 | 13.0 |
| N | 1.6 | 1.0 | 0.7 | 4.2 |
| S | 0.09 | 0.02 | <0.001 | 007 |
| Ash | 68.4 | 83.1 | 3.2 | 42.5 |
| Fixed C | 6.1 | 9.4 | 82.5 | 21.2 |
| Volatile matter | 25.5 | 7.5 | 14.3 | 36.3 |
| pH | 6.8 | 9.5 | 9.7 | 9.1 |
| O/C | 0.46 | 0.07 | 0.03 | 0.26 |
| H/C | 1.29 | 0.60 | 0.32 | 0.89 |
| B. Elemental Analysis of Ash (%, Ash wt Basis) | | | | |
| Al | 3.0 | 2.9 | 0.9 | 0.9 |
| As | <0.005 | <0.005 | 0.1 | <0.005 |
| Ca | 3.0 | 2.8 | 11.8 | 11.6 |
| Cd | <0.005 | <0.005 | <0.005 | <0.005 |
| Cl | <0.01 | <0.01 | 0.6 | 5.6 |
| Cr | <0.005 | <0.005 | 0.15 | 0.01 |
| Cu | 0.005 | 0.005 | 0.26 | 0.4 |
| Fe | 1.43 | 1.41 | 1.13 | 1.11 |
| K | 2.2 | 2.13 | 3.9 | 18.0 |
| Mg | 0.93 | 0.90 | 2.6 | 3.9 |
| Mn | 0.09 | 0.10 | 0.35 | 0.28 |
| Na | 0.31 | 0.30 | 1.1 | 4.5 |
| Ni | 0.005 | 0.006 | 0.03 | 0.016 |
| P | 0.67 | 0.68 | 0.4 | 8.6 |
| Pb | <0.005 | <0.005 | 0.09 | <0.005 |
| S | 0.25 | 0.22 | 0.58 | 4.9 |
| Si | 77.6 | 77.2 | 18.2 | 8.4 |
| Zn | 0.03 | 0.03 | 0.09 | 0.23 |

*2.4. Tissue Analyses for Cadmium and Zinc Concentrations in Shoots and Roots of Corn*

At day 35, corn roots were observed to grow out of the pot bottoms. The experiment was terminated, and the corn shoots and roots were harvested from each pot, oven-dried (60 °C), and digested as described by Hunag and Schulte [31]. Snipped samples were digested in an auto-block using a mixture of nitric and hydrogen peroxide. The concentrations of Cd and Zn in the tissues were analyzed using an ICP spectroscopy. Tissue uptake of Cd and Zn were calculated using Equation (1) for the shoot's uptake and Equation (2) for the root's uptake.

$$MU_{Cd, Zn} = [MBC_{d, Zn}] \times SBY \tag{1}$$

where: MU = metal uptake (kg ha$^{-1}$); CM = concentration of Cd and Zn (%) in corn shoot tissues; SBY = dry matter yield of shoots (kg ha$^{-1}$).

$$MU_{Cd, Zn} = [MBC_{d, Zn}] \times RBY \tag{2}$$

where: MU = metal uptake (kg ha$^{-1}$); CM = concentration of Cd and Zn (%) in corn root tissues; RBY = dry matter yield of roots (kg ha$^{-1}$).

*2.5. Bioconcentration Factor of Cd and Zn in Shoots and Roots of Corn*

The bioconcentration factor (BCF) in corn was calculated as the ratio between heavy metal concentration in the plants (shoots and roots) and the total heavy metal in the soil as shown in Equations (3) and (4).

$$BCF_{shoots} = [MBC_{d, Zn}]_{shoots}/[MBC_{d, Zn}]_{soils} \tag{3}$$

$$BCF_{roots} = [MBC_{d, Zn}]_{roots}/[MBC_{d, Zn}]_{soils} \tag{4}$$

where: $BCF_{roots}$ = bioconcentration factor for Cd and Zn in the roots of corn; $BCF_{shoots}$ = bioconcentration factor for Cd and Zn in the shoots of corn; $CM_{shoot}$ = concentration of Cd and Zn (%) in the corn shoot; and $CM_{soils}$ = concentration of Cd and Zn (%) in the soil.

*2.6. Statistical Analysis*

To determine the effect of different biochar additions (BA) and rates of biochar additions (BR) with or without the manure-based compost (MBC) on biomass and uptake (Cd and Zn) of corn grown in mine soils, data were analyzed with a three-way ANOVA using PROC GLM [32]. For this study, the *F*-test indicated significant results at 5% level of significance, so means of the main treatments (additions of biochars, BA), sub-treatments (rates of biochar additions, BR), sub-sub treatments (rates of MBC, RMBC) were separated following the procedures of the least significance differences (LSD) test, using appropriate mean squares [32].

## 3. Results

*3.1. Soil pH and Water-Soluble Cd and Zn Concentrations in Mine Soils*

Soil pH and concentrations of water-soluble Cd and Zn in mine spoil soils varied significantly with BA ($p \leq 0.0001$), BR ($p \leq 0.0001$), and RMBC ($p \leq 0.0001$). While soil pH was not affected by the interaction effect of BR × RMBC, soil pH and concentrations of Cd and Zn in the soils were significantly affected by the interactions of BA × BR × RMBC (Table 3). Incorporation of 5% PL with 5% RMBC resulted in significantly higher soil pH (6.61 ± 0.01), but significantly lower concentrations of Cd (0.63 ± 0.16 mg kg$^{-1}$) and Zn (10.69 ± 1.95 mg kg$^{-1}$) when compared with the control soils (pH of 4.73 ± 0.32; Cd of 1.89 ± 0.35 mg kg$^{-1}$; Zn of 63.89 ± 11.08 mg kg$^{-1}$).

**Table 3.** Average concentrations of water-soluble Cd and Zn and pH in mine spoil soil.

| Biochar Additions | Biochar Rate (%) | Compost Rate (%) | pH | Cd (mg/kg) | Zn (mg/kg) |
|---|---|---|---|---|---|
| **Control** | **0** | 0 | 4.40 ± 0.06 | 2.05 ± 0.22 | 62.06 ± 6.21 |
| | | 2.5 | 4.69 ± 0.05 | 2.12 ± 0.13 | 70.38 ± 4.20 |
| | | 5.0 | 5.10 ± 0.03 | 1.51 ± 0.08 | 57.12 ± 9.68 |
| | **Mean** | | **4.73 ± 0.32** | **1.89 ± 0.35** | **63.89 ± 11.08** |
| **Beef Cattle Manure** | 2.5 | 0 | 5.07 ± 0.14 | 1.75 ± 0.15 | 56.32 ± 5.06 |
| | | 2.5 | 5.19 ± 0.07 | 1.37 ± 0.11 | 51.11 ± 3.51 |
| | | 5.0 | 5.28 ± 0.12 | 1.10 ± 0.05 | 42.77 ± 2.72 |
| | **Mean** | | **5.18 ± 0.13** | **1.41 ± 0.29** | **49.73 ± 7.22** |
| | 5.0 | 0 | 5.31 ± 0.22 | 1.68 ± 0.14 | 53.81 ± 3.81 |
| | | 2.5 | 5.61 ± 0.14 | 1.04 ± 0.15 | 37.25 ± 4.52 |
| | | 5.0 | 5.91 ± 0.14 | 0.94 ± 0.26 | 32.85 ± 7.84 |
| | **Mean** | | **5.61 ± 0.30** | **1.22 ± 0.39** | **41.31 ± 10.76** |
| **Lodge Pole Pine** | 2.5 | 0 | 4.37 ± 0.01 | 2.57 ± 0.59 | 75.22 ± 7.26 |
| | | 2.5 | 4.77 ± 0.07 | 2.31 ± 0.12 | 75.08 ± 4.69 |
| | | 5.0 | 5.10 ± 0.03 | 1.50 ± 0.04 | 53.27 ± 1.10 |
| | **Mean** | | **4.75 ± 0.26** | **2.13 ± 0.57** | **67.85 ± 6.14** |
| | 5.0 | 0 | 4.47 ± 0.02 | 2.56 ± 0.04 | 70.86 ± 1.96 |
| | | 2.5 | 4.89 ± 0.10 | 1.69 ± 0.32 | 52.35 ± 9.91 |
| | | 5.0 | 5.05 ± 0.05 | 2.04 ± 0.27 | 68.47 ± 9.21 |
| | **Mean** | | **4.81 ± 0.26** | **2.08 ± 0.44** | **63.89 ± 11.08** |
| **Poultry Litter** | 2.5 | 0 | 5.46 ± 0.16 | 3.38 ± 0.89 | 94.02 ± 22.62 |
| | | 2.5 | 5.58 ± 0.24 | 1.94 ± 0.02 | 60.48 ± 6.42 |
| | | 5.0 | 5.85 ± 0.02 | 1.49 ± 0.13 | 47.53 ± 3.42 |
| | **Mean** | | **5.63 ± 0.23** | **2.27 ± 0.98** | **67.35 ± 23.93** |
| | 5.0 | 0 | 6.33 ± 0.03 | 1.19 ± 0.02 | 20.57 ± 1.17 |
| | | 2.5 | 6.53 ± 0.01 | 0.84 ± 0.07 | 13.28 ± 1.08 |
| | | 5.0 | 6.61 ± 0.01 | 0.63 ± 0.16 | 10.69 ± 1.95 |
| | **Mean** | | **6.49 ± 0.13** | **0.89 ± 0.26** | **14.85 ± 4.61** |

| **Sources of Variation** | **Level of Significance** | | |
|---|---|---|---|
| Biochar Additions (BA) | *** | *** | *** |
| Biochar Rate (BR) | *** | *** | *** |
| Compost Rate (RMBC) | ns | *** | *** |
| BA × BR | *** | *** | *** |
| BA × RMBC | ** | ** | *** |
| BR × RMBC | ns | ns | ns |
| BA × BR × RMBC | ns | ** | * |

*** Significant at $p \leq 0.0001$; ** Significant at $p \leq 0.001$; * Significant at $p \leq 0.01$; ns – not significant.

Of the different additions of biochar (BA) when averaged across BR and RMBC, the greatest soil pH increase was from soil treated with PL (6.06 ± 0.18) followed by BCM (5.39 ± 0.21), LPP (4.78 ± 0.26) and control soil (4.73 ± 0.32). The effect of BA on water-soluble Cd (mg kg$^{-1}$) is as follows: LPP (2.10 ± 0.51) > control (1.89 ± 0.35) > PL (1.58 ± 0.62) > BCM (1.32 ± 0.34). The greatest average concentration of water-soluble Zn (mg kg$^{-1}$) was from soil treated with LPP (65.87 ± 8.61) followed by control soil (63.89 ± 11.08), BCM (45.52 ± 8.99), and PL (41.10 ± 28.54) (Table 3).

Overall, the pH of mine soils was significantly affected by the increasing rate (2.5% to 5.0%) of different BA (Table 2). The soil pH of mine soils treated with 2.5% and 5.0% BCM was increased from 5.18 ± 0.13 to 5.61 ± 0.30. Similarly, the pH of soils treated with 2% and 5% LPP was increased from 4.75 ± 0.26 to 4.81 ± 0.26. A much higher increase in the pH of mine soils when treated with 2.5% PL (5.63 ± 0.23) and 5% PL (6.49 ± 0.13). On the other hand, the concentration of water-soluble Cd showed a decreasing trend with the increasing rate of BA application (i.e., 2.5% to 5%). The concentration of water-soluble Cd (mg kg$^{-1}$) in soils was reduced from 1.41 ± 0.29 to 1.22 ± 0.39; 2.13 ± 0.57 to 2.08 ± 0.44; and 2.27 ± 0.89 to 0.89 ± 0.26 when treated with 2.5% and 5% BCM; LPP; and PL, respectively. The concentrations of Cd in the soils were also reduced significantly following the addition of raw beef cattle manure (Table 3). The concentrations of water-soluble Zn (mg kg$^{-1}$) in the soil also showed

decreasing trends following the additions of increasing rates of biochars and beef cattle manure compost. The concentration of water-soluble Zn (mg kg$^{-1}$) in soils was reduced from 49.73 ± 7.22 to 41.31 ± 10.76; 67.85 ± 6.14 to 63.89 ± 11.08; and 67.35 ± 23.93 to 14.85 ± 4.61 when treated with 2.5% and 5% BCM; LPP; and PL, respectively. Again, results have shown the beneficial effects of increasing rates of biochar in combination with the increasing rates application of compost beef cattle manure on enhancing the soil pH while decreasing the concentrations of water-soluble Cd and Zn in mine soils.

*3.2. Concentrations of Cd and Zn in Corn Shoots and Roots*

Except for the concentration of Cd in the shoots, all other concentrations of Cd and Zn in the shoots and roots varied significantly with BA ($p \leq 0.0001$), BR ($p \leq 0.0001$), and RMBC ($p \leq 0.0001$). The interactions of BA × BR and BA × RMBC showed highly significant effects on the Cd and Zn concentrations both in corn shoots and roots (Table 4).

**Table 4.** Average concentrations of Cd and Zn in shoots and roots biomass of corn.

| Biochar Additions | Biochar Rate (%) | Compost Rate (%) | Cd (mg/kg) | Zn (mg/kg) | Cd (mg/kg) | Zn (mg/kg) |
|---|---|---|---|---|---|---|
| | | | **Shoots** | | **Roots** | |
| Control | 0 | 0 | 210.7 ± 49.8 | 3485.3 ± 874.6 | 150.1 ± 29.2 | 3235.2 ± 354.4 |
| | | 2.5 | 145.5 ± 20.9 | 3870.1 ± 512.4 | 255.3 ± 67.2 | 3686.7 ± 801.8 |
| | | 5.0 | 99.1 ± 12.8 | 3165.5 ± 363.6 | 246.9 ± 19.5 | 3531.7 ± 240.2 |
| | **Mean** | | **151.7 ± 55.9** | **3506.9 ± 477.3** | **217.5 ± 62.7** | **3484.5 ± 496.0** |
| Beef Cattle Manure | 2.5 | 0 | 202.8 ± 20.9 | 4881.1 ± 239.3 | 270.9 ± 32.7 | 4390.2 ± 442.9 |
| | | 2.5 | 123.1 ± 17.3 | 3591.3 ± 313.5 | 277.2 ± 31.9 | 3569.1 ± 466.1 |
| | | 5.0 | 96.7 ± 7.1 | 2716.7 ± 151.6 | 245.7 ± 50.4 | 2863.8 ± 211.5 |
| | **Mean** | | **140.8 ± 49.9** | **3729.7 ± 966.3** | **264.6 ± 36.9** | **3607.7 ± 512.8** |
| | 5.0 | 0 | 178.5 ± 7.4 | 4437.5 ± 42.9 | 282.3 ± 44.0 | 3723.2 ± 266.3 |
| | | 2.5 | 99.2 ± 8.3 | 2508.5 ± 282.6 | 216.8 ± 18.8 | 2681.2 ± 158.9 |
| | | 5.0 | 69.1 ± 0.4 | 1575.2 ± 121.6 | 188.7 ± 45.3 | 2053.1 ± 417.6 |
| | **Mean** | | **115.6 ± 49.3** | **2840.4 ± 273.7** | **229.3 ± 42.8** | **2819.2 ± 512.8** |
| Lodge Pole Pine | 2.5 | 0 | 154.4 ± 59.9 | 2611.1 ± 123.9 | 151.2 ± 38.7 | 2666.9 ± 557.3 |
| | | 2.5 | 170.4 ± 26.9 | 4145.5 ± 448.9 | 228.1 ± 74.3 | 3273.6 ± 736.1 |
| | | 5.0 | 155.6 ± 16.9 | 4236.9 ± 618.1 | 229.2 ± 3.0 | 3102.9 ± 194.2 |
| | **Mean** | | **160.1 ± 34.8** | **3664.5 ± 440.4** | **202.9 ± 57.1** | **3014.5 ± 554.0** |
| | 5.0 | 0 | 214.3 ± 42.8 | 3273.8 ± 645.9 | 152.9 ± 16.9 | 2933.0 ± 498.4 |
| | | 2.5 | 167.1 ± 23.2 | 3920.8 ± 340.7 | 172.2 ± 38.1 | 2985.4 ± 432.2 |
| | | 5.0 | 139.8 ± 12.1 | 3577.4 ± 252.6 | 210.3 ± 36.1 | 2850.4 ± 253.9 |
| | **Mean** | | **173.7 ± 41.2** | **3590.7 ± 477.3** | **178.5 ± 37.4** | **2922.9 ± 358.3** |
| Poultry Litter | 2.5 | 0 | 231.4 ± 21.2 | 3127.1 ± 112.9 | 227.9 ± 45.2 | 2222.9 ± 177.9 |
| | | 2.5 | 160.6 ± 13.1 | 2227.8 ± 171.4 | 256.8 ± 77.6 | 2101.7 ± 170.4 |
| | | 5.0 | 126.2 ± 11.6 | 1681.3 ± 157.2 | 159.8 ± 23.7 | 1892.5 ± 287.8 |
| | **Mean** | | **172.7 ± 48.1** | **2345.4 ± 158.9** | **214.9 ± 63.4** | **2072.3 ± 238.4** |
| | 5.0 | 0 | 79.3 ± 17.4 | 651.8 ± 130.5 | 87.8 ± 15.5 | 982.9 ± 158.9 |
| | | 2.5 | 55.4 ± 10.6 | 467.2 ± 72.5 | 51.6 ± 5.4 | 623.3 ± 125.4 |
| | | 5.0 | 50.72 ± 5.7 | 474.8 ± 65.7 | 53.2 ± 5.4 | 655.1 ± 114.1 |
| | **Mean** | | **61.9 ± 16.9** | **531.3 ± 121.8** | **64.2 ± 18.8** | **753.8 ± 116.8** |
| **Sources of Variation** | | | **Level of Significance** | | | |
| Biochar Additions (BA) | | | *** | *** | *** | *** |
| Biochar Rate (BR) | | | *** | *** | *** | *** |
| Compost Rate (RMBC) | | | ns | ** | *** | *** |
| BA × BR | | | *** | *** | *** | *** |
| BA × RMBC | | | ** | *** | ** | *** |
| BR × RMBC | | | ns | ns | ns | ns |
| BA × BR × RMBC | | | ns | ns | ** | * |

*** Significant at $p \leq 0.0001$; ** Significant at $p \leq 0.001$; * Significant at $p \leq 0.01$; ns – not significant.

Overall, the concentrations of Cd and Zn in the shoots and roots with different additions of biochars when averaged across BR and RMBC were significantly lower than the concentrations of Cd and Zn in the shoots and roots of untreated corn. Applications of 2.5% and 5% PL resulted in the

most significant reductions of Cd and Zn concentrations (mg kg$^{-1}$) in the shoots and roots of corn when compared with BCM and LPP with mean values of 172.7 ± 48.1 to 61.9 ± 16.9; 531.3 ± 121.8 to 214.9 ± 63.4; and 2354.4 ± 158.9 to 531.3 ± 121.8; and 2072.3 ± 238.4 to 753.8 ± 116.8, respectively (Table 4). These values were significantly lower than the concentrations of Cd and Zn both in the shoots and roots of untreated corn, suggesting the beneficial effects of biochar applications in phytostabilizing Cd and Zn using corn in mine soils.

### 3.3. Corn Shoots and Roots Biomass

The greatest total corn biomass (kg ha$^{-1}$) was from soils treated with PL (7122.3) followed by BCM (7005.6), and LPP (5008.7). The lowest total biomass of corn was from the untreated soils with a mean value of 5201.6 kg ha$^{-1}$ (Figure 1). The shoot biomass varied significantly with BA ($p \leq 0.0001$) and RMBC ($p \leq 0.0001$), but not with BR (Table 5). On the other hand, the root biomass varied significantly with BA ($p \leq 0.0001$), BR ($p \leq 0.05$), and RMBC ($p \leq 0.05$). The interaction effects of BA × BR × RMBC failed to significantly affect the shoots and roots biomass of corn (Tables 5 and 6).

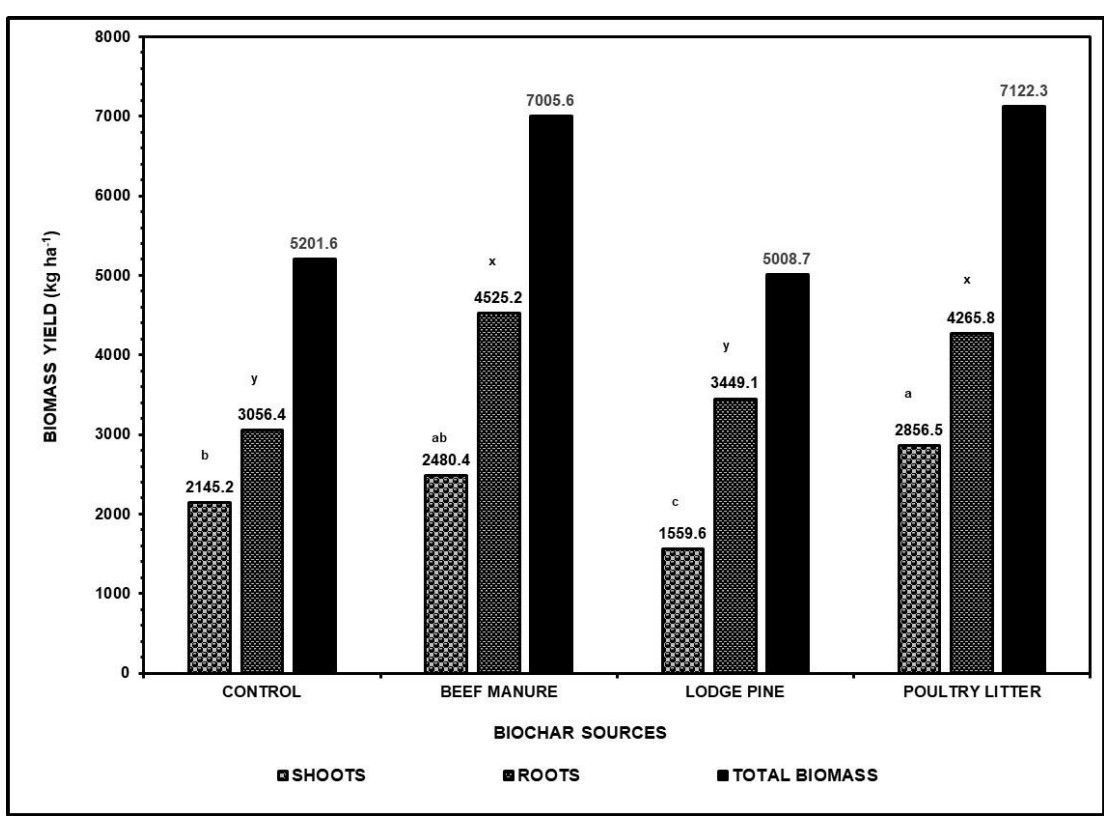

**Figure 1.** Shoots, roots, and total biomass yield of corn applied with different additions of biochars.

**Table 5.** Average shoots biomass (SBY) and uptake of Cd and Zn in shoot biomass of corn.

| Biochar Additions | Biochar Rate (%) | Compost Rate (%) | SBY (kg/ha) | Cd (kg/ha) | Zn (kg/ha) |
|---|---|---|---|---|---|
| | | 0 | 850.3 ± 49.7 | 18.0 ± 4.9 | 298.7 ± 86.4 |
| | 0 | 2.5 | 2119.1 ± 139.5 | 30.7 ± 2.9 | 816.3 ± 74.8 |
| | | 5.0 | 3466.1 ± 711.3 | 33.8 ± 2.9 | 1079.1 ± 103.4 |
| **Control** | **Mean** | | **2145.2 ± 189.6** | **27.5 ± 7.9** | **731.6 ± 352.7** |
| | | 0 | 2020.4 ± 428.6 | 40.4 ± 4.03 | 979.8 ± 161.5 |
| | 2.5 | 2.5 | 2300.4 ± 506.7 | 27.8 ± 2.79 | 817.2 ± 119.3 |
| | | 5.0 | 3544.2 ± 225.7 | 34.2 ± 2.28 | 961.1±40.3 |
| **Beef Cattle Manure** | **Mean** | | **2621.6 ± 785.0** | **34.1 ± 6.1** | **919.4 ± 128.2** |
| | | 0 | 2024.8 ± 380.8 | 35.9 ± 5.4 | 897.9 ± 163.4 |
| | 5.0 | 2.5 | 2524.4 ± 968.9 | 24.6 ± 7.4 | 626.4 ± 215.5 |
| | | 5.0 | 2468.4 ± 623.3 | 17.1 ± 4.4 | 387.5 ± 91.1 |
| | **Mean** | | **2339.2 ± 651.4** | **25.9 ± 9.7** | **637.3 ± 263.2** |
| | | 0 | 781.1 ± 150.5 | 12.6 ± 7.1 | 214.4 ± 127.8 |
| | 2.5 | 2.5 | 1427.9 ± 150.5 | 23.5 ± 6.7 | 579.0 ± 187.4 |
| | | 5.0 | 2220.8 ± 314.9 | 34.2 ± 2.0 | 930.6 ± 74.4 |
| **Lodge Pole Pine** | **Mean** | | **1476.6 ± 702.6** | **23.4 ± 10.6** | **574.6 ± 332.3** |
| | | 0 | 654.3 ± 71.1 | 13.9 ± 3.0 | 212.8 ± 39.0 |
| | 5.0 | 2.5 | 1979.1 ± 248.5 | 32.9 ± 4.2 | 774.4 ± 97.3 |
| | | 5.0 | 2294.5 ± 845.0 | 31.8 ± 10.6 | 819.3 ± 91.8 |
| | **Mean** | | **1642.6 ± 873.7** | **26.2 ± 10.9** | **602.2 ± 331.2** |
| | | 0 | 2368.2 ± 607.5 | 54.2 ± 11.2 | 737.9 ± 174.4 |
| | 2.5 | 2.5 | 3125.7 ± 980.3 | 49.9 ± 14.6 | 689.6 ± 181.7 |
| | | 5.0 | 3187.5 ± 203.1 | 40.2 ± 2.8 | 538.1 ± 82.5 |
| **Poultry Litter** | **Mean** | | **2893.8 ± 706.4** | **48.1 ± 11.2** | **655.2 ± 160.4** |
| | | 0 | 3242.1 ± 861.6 | 25.5 ± 7.3 | 208.4 ± 50.8 |
| | 5.0 | 2.5 | 2766.1 ± 272.5 | 15.1 ± 1.8 | 127.9 ± 9.0 |
| | | 5.0 | 2449.2 ± 433.1 | 12.3 ± 0.9 | 114.7 ± 12.8 |
| | **Mean** | | **2819.1 ± 608.7** | **17.6 ± 7.1** | **150.3 ± 51.3** |

| Sources of Variation | Level of Significance | | |
|---|---|---|---|
| Biochar Additions (BA) | *** | ** | *** |
| Biochar Rate (BR) | ns | *** | *** |
| Compost Rate (RMBC) | *** | ns | *** |
| BA × BR | ns | *** | *** |
| BA × RMBC | ** | *** | * |
| BR × RMBC | * | ns | ** |
| BA × BR × RMBC | ns | ns | ** |

*** Significant at $p \leq 0.0001$; ** Significant at $p \leq 0.001$; * Significant at $p \leq 0.01$; ns – not significant.

**Table 6.** Average roots biomass (RBY) and uptake of Cd and Zn in root biomass of corn.

| Biochar Additions | Biochar Rate (%) | Compost Rate (%) | RBY (kg/ha) | Cd (kg/ha) | Zn (kg/ha) |
|---|---|---|---|---|---|
| Control | 0 | 0 | 2010.7 ± 122.6 | 45.0 ± 7.5 | 972.7 ± 94.1 |
| | | 2.5 | 2738.1 ± 496.6 | 71.5 ± 30.5 | 1025.8 ± 181.8 |
| | | 5.0 | 3420.4 ± 456.9 | 84.2 ± 8.4 | 1202.8 ± 123.7 |
| | Mean | | 3056.4 ± 453.9 | 66.9 ± 23.7 | 1067.1 ± 231.1 |
| Beef Cattle Manure | 2.5 | 0 | 3667.9 ± 414.8 | 99.7 ± 19.2 | 1615.8 ± 293.9 |
| | | 2.5 | 4079.1 ± 592.7 | 111.8 ± 3.7 | 1437.8 ± 56.6 |
| | | 5.0 | 4292.8 ± 719.2 | 104.9 ± 27.0 | 1228.6 ± 224.8 |
| | Mean | | 4013.3 ± 579.5 | 105.5 ± 17.5 | 1427.4 ± 251.4 |
| | 5.0 | 0 | 4211.7 ± 210.1 | 104.9 ± 27.0 | 1570.3 ± 169.6 |
| | | 2.5 | 5570.5 ± 840.5 | 119.5 ± 23.6 | 1493.9 ± 241.6 |
| | | 5.0 | 5328.8 ± 179.7 | 120.2 ± 15.3 | 1101.3 ± 339.8 |
| | Mean | | 5036.9 ± 964.2 | 114.0 ± 25.5 | 1388.5 ± 313.3 |
| Lodge Pole Pine | 2.5 | 0 | 2586.3 ± 180.1 | 39.4 ± 11.9 | 695.1 ± 186.9 |
| | | 2.5 | 2670.3 ± 338.2 | 61.9 ± 26.6 | 887.2 ± 301.6 |
| | | 5.0 | 4723.1 ± 989.8 | 108.4 ± 23.6 | 1473.5 ± 367.1 |
| | Mean | | 3326.6 ± 174.7 | 69.9 ± 35.8 | 1018.6 ± 434.2 |
| | 5.0 | 0 | 2125.0 ± 310.2 | 32.5 ± 6.3 | 631.8 ± 197.2 |
| | | 2.5 | 3547.1 ± 263.2 | 60.4 ± 99.6 | 1051.4 ± 69.5 |
| | | 5.0 | 5042.9 ± 806.2 | 99.6 ± 37.2 | 1394.7 ± 648.8 |
| | Mean | | 3571.7 ± 189.2 | 64.2 ± 34.9 | 1025.9 ± 475.1 |
| Poultry Litter | 2.5 | 0 | 4195.5 ± 864.4 | 93.8 ± 13.9 | 931.3 ± 202.9 |
| | | 2.5 | 3704.8 ± 610.5 | 97.6 ± 40.5 | 783.5 ± 76.7 |
| | | 5.0 | 4141.0 ± 994.6 | 67.4 ± 24.4 | 799.3 ± 298.3 |
| | Mean | | 4013.8 ± 762.6 | 86.3 ± 28.4 | 838.0 ± 212.8 |
| | 5.0 | 0 | 5832.8 ± 604.9 | 52.3 ± 20.5 | 588.3 ± 246.7 |
| | | 2.5 | 3765.2 ± 668.6 | 19.6 ± 5.2 | 236.6 ± 71.4 |
| | | 5.0 | 3955.3 ± 488.8 | 21.1 ± 3.6 | 259.5 ± 36.6 |
| | Mean | | 4517.8 ± 339.7 | 31.1 ± 19.2 | 361.4 ± 214.2 |

| Sources of Variation | Level of Significance | | |
|---|---|---|---|
| Biochar Additions (BA) | *** | *** | *** |
| Biochar Rate (BR) | * | ** | * |
| Compost Rate (RMBC) | * | ns | ns |
| BA × BR | ns | ** | * |
| BA × RMBC | ** | ** | * |
| BR × RMBC | ns | ns | ns |
| BA × BR × RMBC | ns | ns | ns |

*** Significant at $p \leq 0.0001$; ** Significant at $p \leq 0.001$; * Significant at $p \leq 0.01$; ns – not significant.

The effect of BA on the shoot biomass (kg ha$^{-1}$) is as follows: PL (2856.6) > BCM (2480.4) > Control (2145.2) > LPP (1559.6) while the effect of BA on the root biomass is the following: PL (4265.8) > BCM (4525.2) > LPP (3449.1) > Control (3056.4). The mean shoot biomass (kg ha$^{-1}$) of corn following application of 2.5% BCM was about 2621.6 ± 785.0 compared with 2339.2 ± 651.4 from corn treated with 5% BCM. The application of 2.5% LPP and 5% LPP resulted in 1476.6 ± 702 and 1642.6 ± 873.7 while the application of 2.5% PL and 5% PL resulted in 2893.8 ± 706.4 and 2819.1 ± 608.7 kg ha$^{-1}$ of the shoots biomass (Table 5). The effect of increasing rates of the beef manure biochar was more significant because of the increasing trend in the root biomass.

The application of 2.5% LPP and 5% LPP resulted in 3326.6 ± 174.7 and 3571.7 ± 189.2 while the application of 2.5% PL and 5% PL resulted in 413.8 ± 762.6 and 4517.8 ± 339.7 kg ha$^{-1}$ of the roots biomass. The mean corn root biomass (kg ha$^{-1}$) following the application of 2.5% BCM was about 4013.3 ± 579.5 compared with 5036.9 ± 964.2 from corn treated with 5% BCM. These roots biomass following the application of 2.5% and 5% BCM, 2.5% and 5% LPP, and 2.5% and 5% PL were 31.3% and 64.8%, 8.8% and 16.8%, and 31.3% and 47.8% more when compared with the root biomass from the untreated corn plants, respectively (Table 6). Overall, our results show the beneficial effects of

biochars in combination with the compost on enhancing the shoot and root biomass of corn grown in this mine soil.

### 3.4. Uptake and Bioconcentration Factor of Cd and Zn by Shoots and Roots of Corn

Except for LPP, all applications of biochars had significantly enhanced the shoot uptake of Cd and Zn when compared to the Cd and Zn uptake of untreated corn (Table 5). Similarly, all applications of biochar had significantly enhanced the root uptake of Cd and Zn, except for LPP when compared with the Cd and Zn uptake of the control plants (Table 6). Compared to the shoot uptake (kg ha$^{-1}$) of Cd and Zn by the control plants of 18.0 ± 4.9 and 298.7 ± 86.4, the application of BCM, LPP, and PL resulted in an average increase of the Cd shoot uptake of 112.2%, −26.7%, and 121.7% and Zn shoot uptake of 214.3%, −46.3%, and 58.8%, respectively (Table 5). On the root uptake of Cd and Zn, the application of BCM, LPP, and PL resulted in 127.3% and 63.8%, −20.2% and −31.8%, 62.4% and −21.9% over the untreated plants, respectively (Table 6). These results suggest that the effects of biochar application on the shoot and root uptake of Cd and Zn by corn may vary significantly with biochars produced from different feedstocks.

The interaction effects of BA × BR × RMBC did not affect the shoot and root uptake of Cd and Zn by corn (Tables 5 and 6). However, the shoot uptake of Zn by corn varied significantly with the interaction of BA × BR × RMBC. The greatest shoot uptake of Zn was from corn plants treated with 2.5% BCM while the least amount of the Zn shoot uptake was from plants applied with 5% PL in combination with 5% raw beef manure. The shoot and root uptake of Cd and Zn by corn varied significantly with the interaction effects of BA × BR (Tables 5 and 6). The greatest shoot uptake of Cd (48.1 kg ha$^{-1}$) was from plant treated with 2.5% PL while the least amount of the Cd shoot uptake was from plants treated with 5% PL. The application of 5% BCM resulted in the greatest root uptake of Cd (114.1 kg ha$^{-1}$) while the application of 5% PL had the least amount of Cd root uptake of 31.1 kg ha$^{-1}$. Corn plants treated with 2.5% BCM (919.4 kg ha$^{-1}$) had the greatest shoot uptake of Zn while the least Zn shoot uptake by corn was from the application of 5% PL with mean value of 150.3 kg ha$^{-1}$. Similarly, the greatest Zn root uptake of 1427.4 kg ha$^{-1}$ was from corn treated with 2.5% BCM and the least amount of root uptake of Zn was from plants applied with 5% PL with mean uptake of 361.4 kg ha$^{-1}$. Our results suggest that corn is an efficient plant in phytostabilizing Cd and Zn when applied with 2.5% biochar with or without compost.

The bioconcentration factor or BCF of Cd and Zn, which is related to the shoot and root uptake of Cd and Zn as affected by BA and BR as shown in Table 7. When averaged across BR, the greatest BCF for Cd was in the shoot of 92.28 due to the application of BCM and the least BCF was from the application of PL (72.81). The BCF for Zn in the shoot is in the order: BCM (71.88) > LPP (55.10) > PL (35.30). Similarly, both the Cd and Zn BCF in the roots are in the order: BCM (187.80 and 70.39) > LPP (90.54 and 45.08) > PL (83.40 and 40.76), respectively (Table 7). These results suggest a beneficial effect of biochar application in enhancing the phytostabilization capacity of corn roots and shoots for Cd and Zn.

**Table 7.** Bioconcentration factor of Cd and Zn in corn as affected by different biochar additions and rates of biochar application.

| Biochar Additions | Biochar Rate (%) | Cd | Zn | Cd | Zn |
|---|---|---|---|---|---|
| | | **Shoots** | | **Roots** | |
| Beef Cattle Manure Beef | 2.5 | 99.81 | 74.99 | 187.65 | 72.54 |
| | 5.0 | 94.75 | 68.76 | 187.95 | 68.24 |
| **Mean** | | **92.28** | **71.88** | **187.80** | **70.39** |
| Lodge Pole Pine Beef | 2.5 | 75.16 | 54.00 | 95.26 | 44.42 |
| | 5.0 | 83.50 | 56.20 | 85.82 | 45.75 |
| **Mean** | | **79.39** | **55.10** | **90.54** | **45.08** |
| Poultry Litter Beef | 2.5 | 76.07 | 34.82 | 94.67 | 30.77 |
| | 5.0 | 69.55 | 35.78 | 72.13 | 50.75 |
| **Mean** | | *72.81* | *35.30* | *83.40* | *40.76* |

## 4. Discussion

Overall, our results showed that mine spoil remediation can be potentially enhanced by using soil amendments capable of immobilizing metal(loid)s when combined with plant species that are tolerant of high levels of contaminants (Table 1). The incorporation of organic amendments improves the quality of mine soils and makes it possible for vegetation to be established [20,21]. Hossain et al. [24] and Dede et al. [26] have reported that the addition of organic amendments (e.g., biochars, sewage sludge, manures) to soil have promoted the phytoextraction process and improved soil conditions to raise the agronomic values of the soils.

Our results validate the beneficial effects of biochars in combination with the beef cattle manure compost on enhancing the shoot and root biomass and nutritional uptake of corn grown in mine soil with heavy metal contaminations. The greatest total corn biomass was from soils treated with manure-based biochars (PL and BCM) and the least total biomass was from wood-based biochar (LPP) untreated soils. The shoot and root biomass varied significantly with different biochar additions. Results have suggested that biochar applications in mine soils are more likely to influence the biomass, and the effect could be long lasting. Several factors could have had affected the outcome of our study. For instance, differences in the rapidity of decomposition and chemical stability between manure-based and wood-based biochars. In addition, the C:N ratio of the biochars, age of feedstocks, and the degree of disintegration or particle size of the biochars can govern the amount of nutrients released in the soil [33,34]. The C:N ratio of the different biochars that were used in the study are as follows: Poultry litter (8.9) < beef cattle manure (13.8) < lodgepole pine (129.3). Lodgepole pine with wide C:N ratio and low nitrogen content (Table 1) is associated with a slow decay while PL and BCM with narrow C:N ratio and containing higher nitrogen content may undergo rapid mineralization. The profound differences in the C:N ratio of these biochars can explain the striking difference in the decomposition rates, hence faster release of nutrients from these additions to the soils. The rates of mineralization in biochars may have had significant effects on the biomass and nutrient uptake of crop. Our results confirmed the significant effects of different additions of biochars with or without beef cattle manures on biomass productivity and Cd and Zn uptake of corn. As observed in our study, improvements in the corn biomass yield after the biochar addition is often attributed to increased water and nutrient retention, improved biological properties and CEC and improvements in soil pH.

Manure-based biochars, particularly when pyrolyzed at higher temperatures (500 °C and above), have been shown to have strong metal binding capabilities [35]; results which are supported by this study with concentrations of water-soluble Cd and Zn lowest in soils amended with both manure-based biochars (PL and BCM). Concomitantly, additions of PL and BCM resulted in increased total plant biomass yields as compared with the untreated soils and wood-based biochar amendments (PLL). These results are potentially indicative of reduced plant toxicity, though another possibility is that reductions in the available soil of Zn and Cd resulted in reduced stress on soil rhizosphere communities.

Rhizospheric microbial communities provide critical ecosystem services, including nutrient cycling and uptake [36], which result in increased soil fertility. Ippolito et al. [37] previously demonstrated that heavy metal concentrations can have a deleterious effect on microbial community diversity, and additional studies have shown reductions in microbial abundance when faced with increased soil heavy metal concentrations, both of which can negatively impact soil health.

The use of biochar has been investigated for in situ remediation of contaminated lands associated with plants [38,39]. Our results suggest that the incorporation of biochar enhanced phytostabilization of Cd and Zn with concentrations of water-soluble Cd and Zn lowest in soils amended with both manure-based biochars (PL and BCM) while improving the biomass productivity of corn. The biochar application has been shown to be effective in metal immobilization, thereby reducing the bioavailability and phytotoxicity of heavy metals. They also reported that the addition of biochars improve agronomic properties by increasing nutrient availability and microbial activity. The uptake of heavy metals by most plant species decreases in the presence of biochars [40–42]. Further benefits of adding biochars to soil have also been reported; these include the adsorption of dissolved organic carbon [43], increases in soil pH and key soil macro-elements [44], and reductions in trace metals in leachates. Our results support the idea that biochar has proven to be effective at reducing the high concentration of soluble Cd and Zn originating from a contaminated soil and we can now more affirmatively say that sorption is one of the mechanisms by which those metals are retained [45].

The concentrations of water-soluble Cd and Zn in the soil treated with 2.5% and 5% biochars in combination with the increasing beef cattle manure were considerably lower when compared with the control. These results showed effective lowering of Cd and Zn in mine soils after harvesting of corn may well relate to soil pH and phytostabilization of Cd and Zn due to the application of different additions of biochars, especially the manure-based biochar. Sorption of Cd and Zn in biochars can be due to complexation of the heavy metals with different functional groups present in the biochar, such as $Ca^{+2}$ and $Mg^{+2}$ [46], $K^+$, $Na^+$ and S [47], or due to physical adsorption [47]. Some other compounds present in the ash, such as carbonates, phosphates or sulphates [48,49] can also help to stabilize heavy metals by precipitation of these compounds with heavy metals [13].

Overall, the pH of mine soils was significantly affected by the increasing rate (2.5% to 5.0%) of different additions of biochars. The soil pH of mine soil treated with 2.5% and 5.0% BCM was increased from 5.2 to 5.61. Similarly, the pH of soils treated with 2% and 5% LPP was increased slightly from 4.7 to 4.8. A much higher increase in the pH of mine soils with 5% PL (6.5) when compared with the control. The application of biochar in our study increased the soil pH and thus enhanced the phytostabilization of metals and our results agreed with the findings of Park et al. [49] and Zhang et al. [50]. The specific mechanism of metal immobilization in the biochar treatments, with increased soil pH, was likely a result in the formation of precipitates such as $Cd(OH)_2$ and $Zn(OH)_2$. For Cd and Zn, the speciation of which in soil solution is more dominated by free metal ion. Shuman [51] reported that at a pH above eight, chemical precipitation took place and therefore retention of Zn in the soil was due to fixation as a solid phase. Singh and Abrol [52] also concluded that above pH 7.9, pH-pZn curves for different soil systems merged and precipitation reactions were controlling Zn retention.

Metal adsorption in the soil, in addition to pH, organic matter has overriding importance on metal solubility and retention in many soils [53]. Few reports in the literature about soil amendments, such as lime and compost being used to reduce the bioavailability of heavy metals [54]. Biochars can also stabilize heavy metals in soils and thus reduce plant uptake [13]. Addition of soil organic matter in the form of BCM has been recognized as a critical component in the retention of heavy metals in our study. For example, soils treated with 5% BA (PL, BCM, or LPP) when combined with 5% BCM had the lowest concentrations of water-soluble Cd and Zn in the soil. A decreasing trend was noted on the concentrations of water-soluble Cd and Zn in soils with increasing rates of the manure-based compost. The addition of MBC may have enhanced the redistribution of Cd and Zn fractions in the soils and enhanced the phytostabilization and bioavailability of these metals [55]. Our results showed that heavy metal concentrations of Cd and Zn in the plants could be profoundly affected by the amount of plant

available heavy metals in the soil. Additionally, it is possible that the increase in soil pH caused by the biochar application could have had enhanced the adsorption and complexation of Cd and Zn on biochar, which caused a decrease in water-soluble Cd and Zn in the soil at 5% level of biochars in our study. It has been shown that organic materials can strongly bind heavy metals such as Cu, Pb, Cd, Zn, and Ni. The solubility of the metals depends mainly on the metal loading over soil sorbents, pH, and the concentration of dissolved organic matter in the soil solution [56].

Another important part of this study is on the effect of different additions and application rates of biochars on the bioconcentration factor (BCF) of Cd and Zn in corn shoots and roots. Plant's ability to accumulate metals from soils can be estimated using BCF, which is defined as the ratio of metal concentration in the shoots or roots to that in the soil. The plant's ability to translocate metals from the roots to the shoots is measured using the translocation factor (TF), which is defined as the ratio of the metal concentration in the shoots to the roots. As shown in our data (Table 7), corn has demonstrated a high degree of tolerance factor because we did not see restriction in soil-root and root-shoot transfers. Corn grown in contaminated mine soils can be considered as a hyperaccumulator because it has actively taken up and translocated Cd and Zn into their biomass. Our results showed that BCF of Cd and Zn varied significantly with the different additions and application rates of biochars. Corn applied with 2.5% BCM has the greatest Cd and Zn BCF in the shoots and these results suggest that corn can accumulate large quantities of metal in their shoot tissues when grown in contaminated mine soils. Based on averaged BCF in corn with different additions and rates of biochars, corn can be considered a minor accumulator of Cd and Zn. However, the BCF values of Cd and Zn in corn (Table 7) were much greater than one, are evident that Cd and Zn in mine soils were highly bio-accumulated and phytostablized. Lu et al. [57] from their study on the removal of Cd and Zn by water hyacinth suggested that water hyacinth as a moderate accumulator of Cd and Zn with BCF values of 622 and 789, respectively. Another study on the use of biochar and phytostabilization using *Brassica napus* L. was conducted to target Cd-polluted soils [7]. Additionally, the results of Hartley et al. [58] and Case et al. [59] showed that biochar can be used in combination with Miscanthus for phytostabilization of Cd and Zn in contaminated soils. Novak et al. [60] from their most recent study on using blends of compost and biochars concluded that the designer biochar is an important management component in developing successful mine site phytostabilization program.

## 5. Summary and Conclusions

In our study, we evaluated the interactive effects of manure- and plant-based biochar applications with or without compost on the shoots and roots biomass production, uptake, and BCF of Zn and Cd of corn grown in mine soil. Results of our study can be summarized as follows:

1.  With increasing rates of biochar in combination with increasing rates the application of manure-based compost enhanced soil pH and decreased the concentrations of water-soluble Cd and Zn in mine soils;
2.  Effects of the biochar application on the shoot and root uptake of Cd and Zn by corn varied significantly with biochars produced from different feedstocks; and
3.  The BCF values of Cd and Zn in corn were considerably greater than one, which are evident that Cd and Zn in mine soils were highly bio-accumulated and phytostablized due to biochar and phytostabilization using corn.

Overall, our results suggest that phytostabilization when combined with the biochar and manure-based compost application have the potential for the remediation of heavy metals polluted soils.

**Author Contributions:** All authors contributed to this research project. Individual contributions to the following categories are as follows: Research Conceptualization: G.C.S., J.M.N., M.G.J., J.I., T.D.D., and K.S.; Methodology: J.M.N., G.C.S., T.D.D., and D.W.W.; Data Analysis: G.C.S.; Writing—Original draft preparation: G.C.S.; Review and editing: J.M.N., J.I., M.G.J., K.S., T.D.D., and D.W.

**Funding:** This research was funded through an Interagency Agreement between the United States Department of Agriculture-Agricultural Research Service (60-6657-1-204) and the United States Environmental Protection Agency (DE-12-92342301-2).

**Acknowledgments:** Gratitude is expressed to the staff of the ARS especially Mr. William Myers, the US EPA locations, and team at the Webb City, MO water treatment facility for their work and diligence with sample collection, preparation and analyses. This work was made possible through an Interagency Agreement between the United States Department of Agriculture-Agricultural Research Service (60-6657-1-204) and the US EPA (DE-12-92342301-2). Approval does not signify that the contents reflect the views of the USDA-ARS or the US EPA, nor does mention of trade names or commercial products constitute endorsement or recommendation for their use. USDA is an equal opportunity provider and employer.

**Conflicts of Interest:** There is no conflict of interests.

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
