# Peer review of "Phytostabilization of Zn and Cd in Mine Soil Using Corn in Combination with Biochars and Manure-Based Compost"

_environments, doi:10.3390/environments6060069_

Round 1

Reviewer 1 Report

The manuscript describes the Cd and Zn concentration in corn after planting it in a mine soil, and studies the positive effects of plant growth due to manure-based char, manure-based compost and the combination of both. The manuscript is well written but some adjustments are needed before publication. The methods are limited which make it difficult to appreciate if the combination of the compost and char really make a difference in real world. The limitations of the methods make the discussion very long. This can be improved by skipping all figures as the data have already been presented in the tables.  The authors seem to have used an extreme soil with very high cadmium concentrations and it is therefore very strange that the authors seem to have forgotten to present data about the soil. Maybe because it has been presented in other studies but no reference has been made to this.

If major adjustments are made the research is still rather restricted. One cannot distinguish why the animal manure-based compost or animal manure-based char have an effect because the compost and the char have various effects on the soil and plants via the binding capacity, nutrients and liming effect. In a more scientific approach lime, or pH buffering and organic material without nutrients, or the addition of a constant amount of nutrients would have been used. This puts strong restictions on the type of conclusions. The authors cope by making a very extended discussion (I would have preferred a better research).

The authors conclude that "phytostabilization and biochar application have the potential to be combined in the remediation of heavy metal polluted soils". For example, it is unclear for the reader if the same effect can be obtained much more easily by using standard lime and fertilizers or compost. I would like to urge the authors to improve the text by making the discussion much shorter: please do not discuss everything that you have not studied.

Detailed comments

4 "compost": cattle manure compost is used in this study. The use of the term "compost" for this aged manure is new to me and I guess also for non-American readers. Many people assume that compost is always plant material, and legislation in Europe is very strict on this. I would prefer the use of the term animal together with compost, or other terms: manure-based compost, or simply animal manure.   

78-86. Why was corn chosen? The plant material has an extreme Cd content and cannot be used for animal feed. Why not use a native plant or a more practical plant for this mining area in Missouri?

107-115. The C-horizon material has not been analyzed for heavy metals, nutrients, organic matter, toxicity etc. Why not? This makes it impossible to compare it to other studies. It is also strange as the problem of the study is given in line 64: "mine soils  can have unfavourable soil chemical characteristics". I think it is impossible to publish this material without mentioning at least the total Cd content of the C-material.

The text gives water-soluble Cd and Zn. The methods mention: water, CaCl2, and DTPA extraction. There is no appropriate description or reference of these methods.

104-107." < 12.7-cm". I assume you mean 12.7 millimeters and not cm?

129. The method of analysis is not appropriate to determine Cd and Pb in compost or biochar. One should be able to test if the materials comply with rules set for compost and biochar. 

115. Please classify if ICP -AES or ICP-MS was used.

Figure 1-5.  The figures are unclear, you cannot read it independent from the text. It should mentioned that the results are an average of 0, 2.5 and 5% manure-based compost.  Please skip all the figures as all results are given in the tables.

Author Response

Reviewer 1 - Comments and Suggestions for Authors

The manuscript describes the Cd and Zn concentration in corn after planting it in a mine soil, and studies the positive effects of plant growth due to manure-based char, manure-based compost and the combination of both. The manuscript is well written, but some adjustments are needed before publication. The methods are limited which make it difficult to appreciate if the combination of the compost and char really make a difference in real world. The limitations of the methods make the discussion very long. This can be improved by skipping all figures as the data have already been presented in the tables.  The authors seem to have used an extreme soil with very high cadmium concentrations and it is therefore very strange that the authors seem to have forgotten to present data about the soil. Maybe because it has been presented in other studies, but no reference has been made to this.

If major adjustments are made the research is still rather restricted. One cannot distinguish why the animal manure-based compost or animal manure-based char have an effect because the compost and the char have various effects on the soil and plants via the binding capacity, nutrients and liming effect. In a more scientific approach lime, or pH buffering and organic material without nutrients, or the addition of a constant amount of nutrients would have been used. This puts strong restrictions on the type of conclusions. The authors cope by making a very extended discussion (I would have preferred a better research).

Response:

Both the added manure-based biochar and manure-based compost would enhance the soil organic matter content of the soil. The authors have made several discussion describing remediation of mine soils using biochars and other soil amendments as well as the importance of phytostabilization using plants. Examples of extended discussion are given below:

Numerous studies may have had shown that adding organic amendments (e.g., biochars, sewage sludge, manures) to soil promotes phytoextraction process, but only few studies have evaluated the combined effect of organic amendments and phytostabilization with corn in Cd and Zn contaminated mine soils. There is a lack of agreement over the influence of organic amendments such as biochars on metal immobilization in soil. Moreover, application of biochars to contaminated soil systems has not been systematically investigated to any great extent. Biochar may be a tool for mine spoil remediation; however, its mechanisms for achieving this goal are still not well understood.

Phytostabilization can be enhanced by using soil amendments that immobilize metal(loid)s when combined with plant species that are tolerant of high levels of contaminants while simultaneously improving the physical, chemical, and biological properties of mine soils. Some previously used amendments to improve soil conditions include biosolids, lime, green waste, or biochars. Among these amendment types, the use of biochar has recently been investigated for in situ remediation of contaminated lands in association with plants. The incorporation of organic amendments improves the quality of mine soils and makes it possible for vegetation to be established. Out studies have highlighted that biochars are effective soil amendments in that they improve soil conditions to raise the agronomic values of soils.

Overall, we have revised the paper. As suggested by the reviewer, we revised the materials and methods section, results and discussion section and the summary and conclusion section.

The authors conclude that "phytostabilization and biochar application have the potential to be combined in the remediation of heavy metal polluted soils". For example, it is unclear for the reader if the same effect can be obtained much more easily by using standard lime and fertilizers or compost. I would like to urge the authors to improve the text by making the discussion much shorter: please do not discuss everything that you have not studied.

 Response:

To avoid confusion and make our point much clearer in conveying the overall results of our study on the combined effect of biochar additions with or without manure-based compost, we revised our conclusion as shown below:

In our study, we evaluated the interactive effects of manure- and plant-based biochar applications with or without compost on shoots and roots biomass production, uptake, and BCF of Zn and Cd of corn grown in mine soil. Results of our study can be summarized as follows:

1. with increasing rates of biochar in combination with increasing rates application of manure-based compost enhanced soil pH and decreased the concentrations of water-soluble Cd and Zn in mine soils;

2. effects of biochar application on shoot and root uptake of Cd and Zn by corn varied significantly with biochars produced from different feedstocks; and

3. the BCF values of Cd and Zn in corn were considerably greater than 1, which are evident that Cd and Zn in mine soils were highly bio-accumulated and phytostablized due to biochar and phytostabilization using corn.

Overall, our results suggest that phytostabilization when combined with biochar and manure-based compost application have the potential for the remediation of heavy metals polluted soils.

 Detailed comments

4 "compost": cattle manure compost is used in this study. The use of the term "compost" for this aged manure is new to me and I guess also for non-American readers. Many people assume that compost is always plant material, and legislation in Europe is very strict on this. I would prefer the use of the term animal together with compost, or other terms: manure-based compost, or simply animal manure.   

Response:

As suggested by the reviewer, manure-based compost abbreviated as MBC was used in the manuscript. The title of the paper was revised to emphasize the additions of biochars and manure-based compost. The revised title now is:

Phytostabilization of Zn and Cd in Mine Soil Using Corn in Combination with Biochars and Manure-Based Compost

78-86. Why was corn chosen? The plant material has an extreme Cd content and cannot be used for animal feed. Why not use a native plant or a more practical plant for this mining area in Missouri?

Response:

We have an early paper titled “Biochar compost blends facilitates switchgrass growth in mine soils by reducing Cd and Zn bioavailability” published in Biochar journal (Reference #60). This paper was focused on a native grass in Missouri. The three biochars and compost mixtures have improved switchgrass productivity and could be potential in developing a mine site phytostabilization program. Based on our observation, our biochar treatments with or without manure-based compost may work with energy crop like corn or sorghum. In our study, we considered corn because we wanted to understand if non-grass specie can be tolerant to high concentrations of metals, especially Cd and Zn. We also wanting to know on how can increase the production of corn and how corn can immobilize heavy metals in the soils. The knowledge that we gained from our results can provide new possibilities for solving problems with contamination if water and soils, which are facing around the world.  

Our results showed that phytostabilization technique using corn and biochar additions have the potential to be combined in the remediation of heavy metals polluted soils.

107-115. The C-horizon material has not been analyzed for heavy metals, nutrients, organic matter, toxicity etc. Why not? This makes it impossible to compare it to other studies. It is also strange as the problem of the study is given in line 64: "mine soils can have unfavourable soil chemical characteristics". I think it is impossible to publish this material without mentioning at least the total Cd content of the C-material.

Response:

The soil chemical analyses of the C-horizon materials are shown in Table 1 (added into the revised manuscript, page 4). This table was included in the original draft of the manuscript and was accidentally deleted form the submitted version of the paper. The total metal concentration of Zn in the soil was about 2225 mg/kg; water-soluble concentration of about 141 mg/kg and extractable concentration of 782 mg/kg. Cd concentrations in the soils were: total metal of 72.2; water-soluble of 5.73 and extractable concentration of 50.45 mg/kg. These are the initial concentrations of Zn and Cd in the soil prior to planting of corn.

For line 64, the word “can” was replaced by the word “may” to avoid the strange problem right way when dealing with mine soils.

The text gives water-soluble Cd and Zn. The methods mention: water, CaCl2, and DTPA extraction. There is no appropriate description or reference of these methods.

Response:

The different refences were added into the revised manuscript.

104-107." < 12.7-cm". I assume you mean 12.7 millimeters and not cm?

Response:

Yes, it should be 12.7 mm not cm. Thanks much for the correction. It was corrected in the revised manuscript.

129. The method of analysis is not appropriate to determine Cd and Pb in compost or biochar. One should be able to test if the materials comply with rules set for compost and biochar. 

Response:

The appropriateness was based on a previously published paper where we compare our results. This is what was added in to the revised manuscript.

The appropriateness of using the different designer biochars in our study were based on an early published paper by Novak et al. [16].

115. Please classify if ICP -AES or ICP-MS was used.

Response:

 It was the ICP-AES, this is already corrected in the manuscript, as shown below:

All water-soluble and extractable metal concentrations including Cd and Zn were quantified via Inductively Coupled Plasma spectroscopy atomic emission spectroscopy (ICP-AES). Concentrations of Cd and Zn and other chemical properties of C horizon are presented in Table 1.

Figure 1-5.  The figures are unclear, you cannot read it independent from the text. It should mention that the results are an average of 0, 2.5 and 5% manure-based compost.  Please skip all the figures as all results are given in the tables.

Response:

We agree with the reviewer – all the figures, except for Figure 1 were deleted.

Reviewer 2 Report

-       Introduction: please better introduce the objectives and the layout of the work in the last sentences

-       Lines 181-184: CR, BR must be defined better; it is not clear whether compost addition was always made or not; beef cattle manure was always added, apart from control. Please improve clarity of this paragraph.

-       BS is not a quantitative item, please consider it throughout the manuscript (e.g. interaction of BS…addition of biochar, so it would be better BA, as biochar addition)

-       Lines 194-195: It is not completely true, only at selected rates. Please clarify better this point.

-       Lines 323-325: what is the meaning of any single reference?

-       Lines 415-421: amount of metals bound to Organic Matter is cited, but not measured. Did the authors perform any sequential extraction of both the untreated and the treated soil, to assess metals amount in the different soil fractions (exchangeable, bound to organics, carbonate...)? This could help to explain why decreasing metal bioavailability resulted in an increase of metal uptake (that appear to be in contrast).

-       Conclusions: the last sentence is unnecessary (it is point 1).

Author Response

Reviewer 2 -Comments and Suggestions for Authors

-       Introduction: please better introduce the objectives and the layout of the work in the last sentences

Response:

As suggested by the reviewer, the objective of the study was added into the introduction section as shown below. The introduction section was also slightly revised.

The objective of our study was to evaluate the interactive effects of biochar additions with or without manure-based compost (MBC) on shoots biomass (SBY), roots biomass (RBY), uptake, and bioconcentration factor (BCF) of Zn and Cd in corn (Zea mays L.) grown in mine soil.

-       Lines 181-184: CR, BR must be defined better; it is not clear whether compost addition was always made or not; beef cattle manure was always added, apart from control. Please improve clarity of this paragraph.

Response:

This section was totally revised, see the revision below:

The experimental treatments were consisted of biochar additions (BA): beef cattle manure (BCM); poultry litter (PL); and lodge pole pine (LPP). Biochar additions were applied at 0, 2.5, and 5.0% (w/w) with or without MBC. The different rates of MBC were 0, 2.5, and 5.0% (w/w). Experimental treatments were replicated three times using a 3 x 2 x 3 split plot arrangement in completely randomized block design.

-       BS is not a quantitative item, please consider it throughout the manuscript (e.g. interaction of BS…addition of biochar, so it would be better BA, as biochar addition)

Response:

As suggested by the reviewer, BA as biochar additions was adopted throughout the manuscript including the interaction effects.

-       Lines 194-195: It is not completely true, only at selected rates. Please clarify better this point.

Response:

We agree with the reviewer. To avoid confusion Lines 194-195 were deleted without affecting the results of the study

-       Lines 323-325: what is the meaning of any single reference?

Response:

This line, specifically” any single reference” does not exist in the original submission .

-       Lines 415-421: amount of metals bound to Organic Matter is cited, but not measured. Did the authors perform any sequential extraction of both the untreated and the treated soil, to assess metals amount in the different soil fractions (exchangeable, bound to organics, carbonate...)? This could help to explain why decreasing metal bioavailability resulted in an increase of metal uptake (that appear to be in contrast).

Response:

We did not perform any sequential extraction of the untreated and treated soils, but we have the analyses of the metals in the soils before and after the completion of the study. These data are shown in Table 1 and Table 3, respectively. Results showed a significant reduction in the concentration of Cd and Zn, specially for soils treated with biochars with manure-based compost. The sentence “Organic matter contains S, O, and N … was deleted. These components were not measured in the study, so we agree with the reviewer. To avoid confusion, we deleted the whole sentence.

-       Conclusions: the last sentence is unnecessary (it is point 1).

Response:

As suggested by the reviewer, the conclusion section was revised. The last sentence was deleted. The last sentence in the revised manuscript is shown below:

Overall, our results suggest that phytostabilization when combined with biochar and manure-based compost application have the potential for the remediation of heavy metals polluted soils.

Round 2

Reviewer 1 Report

The manuscript has been improved according to the comments, and much of the problems I had with the manuscript seem to have been due to a missing table. 

Author Response

The manuscript has been improved according to the comments, and much of the problems I had with the manuscript seem to have been due to a missing table. 

Response:

All the tables are all accounted for and properly cited on the text. All the table numbers were highlighted in yellow. The corresponding table was also highlighted in yellow. The revised manuscript has a total of 7 tables and they are all properly cited. For the reviewer, thanks so much for your time reviewing our paper.

Reviewer 2 Report

Lines 323-325: what is the meaning of any single reference?

Please avoid lump references and report the added value of each of them

Author Response

Lines 323-325: what is the meaning of any single reference? 

Please avoid lump references and report the added value of each of them

Response:

Lines 323-325 were the old lines that appeared on our first submission. In the revised manuscript, these lines are now lines 343-346. As suggested by the reviewer, lumping references was avoided. Individual contribution was added into the revised manuscript. See below.

Hossain et al. [24] and Dede et al. [26] have reported that addition of organic amendments (e.g., biochars, sewage sludge, manures) to soil have promoted phytoextraction process and improve soil conditions to raise the agronomic values of the soils.